# POLICY FILTRATION IN RLHF TO FINE-TUNE LLM FOR CODE GENERATION

## ABSTRACT

While direct policy optimization methods exist, pioneering LLMs are fine-tuned with reinforcement learning from human feedback (RLHF) to generate better responses under the supervision of a reward model learned from preference data. One major challenge of RLHF is the inaccuracy of the intermediate reward model, especially in code generation tasks that requires complex reasoning for the reward model to score a response. We find that the reliability of the reward model varies across responses assigned with different rewards. This motivates us to filter the samples whose rewards may be unreliable to improve the signal-to-noise ratio during policy learning, resulting in Policy Filtration for Proximal Policy Optimization (PF-PPO). To choose a proper policy filtering strategy, we use the coefficient of determination ($R^2$) between the rewards and actual scores on filtered samples as the metrics to help us find promising strategies since it measures how well the rewards filtered by PF-PPO indicate real performance. We provide extensive experiments to validate the effectiveness of PF-PPO in code generation tasks. We find that some variants of PF-PPO are highly effective and achieve the state-of-the-art performance of 7-billion-parameter models on HumanEval (+7.9%) and MBPP (+0.7%). Moreover, we create the LeetCode Contest benchmark and demonstrate the advantage of PF-PPO (+10.0%) on this more challenging benchmark.

## 1 INTRODUCTION

Reinforcement Learning from Human Feedback (RLHF) is a key technique to align large language models (LLMs) with human values and preferences (Christiano et al., 2017; Ziegler et al., 2019; Ouyang et al., 2022). RLHF has been proven to be an essential process for LLMs to produce more helpful, harmless, and honest responses (Bai et al., 2022). Despite various non-RL algorithms such as DPO (Rafailov et al., 2024) are proposed, state-of-the-art applications such as ChatGPT/GPT-4 (OpenAI, 2023), Claude (Anthropic, 2023), and Gemini (Team et al., 2023) adopt the RL algorithm (e.g., PPO) for policy optimization. The key challenge of RLHF is the inaccuracy of the intermediate reward model. While there are researchers investigate how to learn reliable reward models (see e.g., Wang et al., 2024), we focus on how to learn better policy under the guidance of such inaccurate reward models.

We observe that, though the reward model gives inaccurate rewards in general, it can be more reliable in specific regions (e.g., when it gives high rewards) than the others. The observation is based on the simple experiment: We use a policy model fine-tuned for code generation to generate a set of responses for prompts in the HumanEval dataset. Later, we score these responses using a reward model trained with the common recipe (see Ouyang et al., 2022, and also Section 2) and compare them with the actual scores. We find that, across different sets of samples, the reward model is more reliable when it gives high or low rewards than when it gives moderate rewards. (This property also holds on other datasets and see Appendix A for more experiment results and futher discussion.) Considering that RLHF updates the policy solely based on the reward signal, this observation motivates us to filter out the samples with possibly unreliable rewards aiming to improve RLHF by increasing the signal-to-noise ratio on training samples.

Based on this motivation, we propose a simple modification to the standard PPO-based RLHF algorithm (Ouyang et al., 2022), Policy Filtration for PPO (PF-PPO), that learns a filtered version of the policy using PPO. Specifically, we generate $N$ samples for each prompt, score these samples

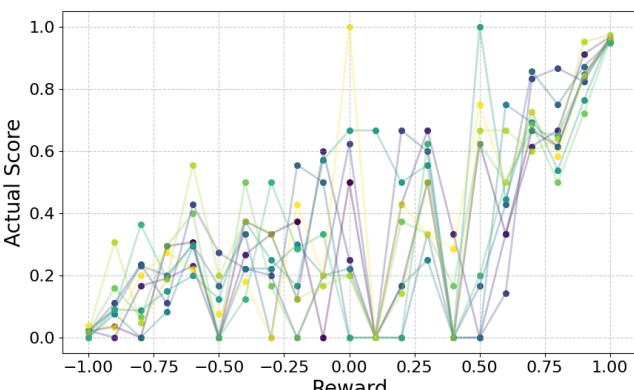

Figure 1: The reward model can be *inaccurate*, i.e., the actual score of the response does not align well with the reward given by the reward model. However, the reward model in specific regions (e.g., when it gives rewards higher than 0.8) is more *reliable*, i.e., the responses with similar rewards result in consistent performance. We use a fine-tuned policy to generate 10 responses for each of the 164 prompts in the HumanEval dataset and use a reward model trained with the common recipe to generate their rewards. We group the responses with similar rewards and calculate the average of their actual scores (i.e., the average correctness), indicating each group by one point. To evaluate the reliability of the reward model, we repeat the process ten times corresponding to the ten lines.

using the reward model, and use a filtered subset of these samples for subsequent policy training. We design filtering strategies to improve the reliability of the reward model on the filtered samples by maximizing the coefficient of determination ($R^2$) between the rewards and actual scores on these filtered samples. We show that the reward model can evaluate more accurately on these filtered samples, thus providing better training signal and improving the performance of the policy. Our method is also connected with reject sampling that filters out responses with low rewards during inference to yield a better response. Reject sampling is a simple but surprisingly strong inference-time strategy, whereas we adopt similar filtration in an RL algorithm.

Empirically, we show that PF-PPO can greatly improve the performance of LLMs on code generation tasks, which is challenging since complex logic behind these tasks makes the reward model inaccurate in general. We conduct extensive ablation studies to validate the design of our algorithm. Moreover, we illustrate the effectiveness of our algorithm by fine-tuning LLMs that achieves new sota on HumanEval and LeetCode Contest benchmarks across 7-billion-parameter LLMs. To evaluate whether PF-PPO can be effective on more challenging coding tasks, we create the LeetCode Contest benchmark that includes competition-level coding tasks for human experts. We find that the policy filtration technique can result in even more significant improvement on this challenging benchmark.

## 2   RELATED WORK

**Limitation of reward model.** The outcome of RLHF highly relies on the quality of the reward model. Unfortunately, the reward model can hardly provide accurate scores due to 1) the mis-specified reward modeling to represent human preferences (Lambert et al., 2023; Pitis, 2023); 2) the presence of incorrect and ambiguous preferences in the dataset (Ouyang et al., 2022; Bai et al., 2022), and 3) the poor generalization ability of the reward model (McKinney et al., 2023). The inaccuracy of reward model is attributed as one major cause of *reward hacking* and *hallucination* in LLMs (Kalai & Vempala, 2024). While there are previous papers try to improve the accuracy of the reward model itself (Wang et al., 2024; Coste et al., 2023; Zhang et al., 2024), the objective of our paper is to design a better RLHF algorithm in the face of inaccurate reward models.

**Reject sampling.** Reject sampling (or best-of-N sampling) is a popular and effective inference-time strategy to enhance the response of an LLM by generating $N$ responses and select the best one according to a reward model (Nakano et al., 2021; Cobbe et al., 2021). This trick can yield good

responses while keeping a tight KL constraint to the original policy. Inspired by its effectiveness in inference, researchers also try to involve this trick in policy optimization. For example, RAFT (Dong et al., 2023), BOND (Sessa et al., 2024) and vBoN (Amini et al., 2024) learn a policy that distills the best-of-$N$ policy using supervised fine-tuning losses. In a boarder sense, the rank information of the $N$ samples can also be leveraged. For example, RRHF (Yuan et al., 2023) and PRO (Song et al., 2024) train the policy using the combination of a ranking loss and a SFT loss (w.r.t. the best response) based on $N$ responses for each prompt. However, these algorithms do not adopt an elaborate RL algorithm, while state-of-the-art language models adopts RL algorithms in alignment, benefiting from the generalization power of the reward model especially in reasoning tasks (Ivison et al., 2024). Unlike these algorithms, we adopt the idea of reject sampling in the sampling phase of an RL algorithm instead of using supervised learning losses.

**RLHF algorithms in the face of inaccurate reward models.** One key challenge in RLHF is the inaccuracy of reward model, which can lead to reward over-optimization (Gao et al., 2023; Skalse et al., 2022; Chaudhari et al., 2024). Optimization with a policy constraint (e.g., a KL divergence between the target policy and the reference policy) is a remedy frequently used in not only RL-based algorithms (Ouyang et al., 2022; Wu et al., 2023; Zhu et al., 2023) but also direct policy optimization algorithms (Rafailov et al., 2024; Zhao et al., 2023; Liu et al., 2023). Going beyond policy constraint, Moskovitz et al. (2023) only maximize rewards up to a threshold to avoid excessive deviation from a pre-trained policy. In this paper, we not only rely on the policy constraint to optimize in the face of inaccurate rewards but also try to avoid using samples with unreliable rewards.

# 3 PRELIMINARY

**Notations.** We use $[a, b]$ to denote the set $\{a, a + 1, \cdots, b\}$ and use $[b]$ as the shorthand for $[1, b]$. We use $\oplus$ to denote the concatenation on tokens, and use $x_{a:b}$ as the shorthand for the concatenation $(x_a \oplus x_{a+1} \oplus \cdots \oplus x_b)$. We use $c_i$ and $y_i$ to indicate the $i$-th token in the context $c$ (including task instruction, prompt, inputs, etc.) and the response $y$ respectively.

**MDP formulation.** We adopt a Markov decision process (MDP) formulation for RLHF. Specifically, language generation is formulated as an MDP $M = (\mathcal{S}, \mathcal{A}, P, R)$ with states $s \in \mathcal{S}$, actions $a \in \mathcal{A}$, transition probabilities $P \in \Delta(\mathcal{S})^{\mathcal{S} \times \mathcal{A}}$, and the next-state-based reward function $R : \mathcal{S} \to [0, 1]$. Given a context $c$ with $T_c$ tokens, on each step $t \in [T_c + 1, T]^1$, the language model $\pi_\theta(a_t | s_t)$ selects a token $a_t = y_{t-T_c}$ based on the state $s_t := (c_{1:T_c} \oplus y_{1:t-T_c-1})$. Then, the language model enters the next state $s_{t+1} := (c_{1:T_c} \oplus y_{1:t-T_c})$ until the language model completes the response $y_{1:T-T_c}$. For simplicity, we will also use contextual-bandit-style notations, e.g., we denote the language generation process as $y \sim \pi_\theta(\cdot | c)$.

**RLHF.** Reinforcement learning with human feedback (RLHF) is an important process to address *objective mismatch* between the next-token-prediction objective in pre-training and our expectation of LLMs to follow the instructions and assist humans to complete various tasks. We briefly review the pipeline of RLHF.

- **Supervised fine-tuning.** In the supervised fine-tuning (SFT) phase, a pre-trained LLM is fine-tuned with a high-quality supervised dataset collected for specific downstream tasks. Typically, the LLM is fine-tuned with a maximum likelihood loss, and we denote the output of this phase as $\pi^{\text{SFT}}$. While subsequent RLHF procedure is necessary for training high-quality LLMs, this phase alone can also yield an LLM that reasonably follows human instructions (see e.g., Longpre et al., 2023).

- **Reward model learning.** In the reward model learning phase, we learn a reward model $r_\phi(y|c) \in [-1, 1]$ parameterized by $\phi$ that scores the response $y$ to the context $c$ based on collected preference data $\mathcal{D}_{\text{HF}} := \{(c, y^w, y^l)\}$ specifying that $y^w$ is a preferred response to $c$ than $y^l$. The reward model is initialized by $\pi^{\text{SFT}}$ with an additional output layer. A preference model links the reward model with the preference data, and Bradley-Terry model (Bradley & Terry, 1952) is a common choice:

$$\mathbb{P}(y^w \succ y^l | c) = \sigma(R_\phi(y^w | c) - R_\phi(y^l | c)), \tag{1}$$

---

[1]We fix the index of the terminal state to be the maximum length $T$. To adapt responses of different lengths, we left pad the context $c$.

where $\sigma$ is the sigmoid function. The learning objective of reward model is to maximize the log-probability on preference data:

$$\max_{\phi} \mathbb{E}_{(c,y_w,y_l)\sim\mathcal{D}_{\text{HF}}} \left[\log \mathbb{P}(y_w \succ y_l|c)\right]. \tag{2}$$

- **RL fine-tuning.** In this stage, we fine-tune the language model $\pi_\theta$ to maximize the rewards given by the reward model with a policy constraint. The optimization problem is formulated as

$$\max_{\theta} \mathbb{E}_c \mathbb{E}_{y\sim\pi_\theta(\cdot|c)} \left[r_\phi(y|c) - \beta D_{\text{KL}}(\pi_\theta(\cdot|c)||\pi^{\text{SFT}}(\cdot|c))\right]. \tag{3}$$

The second term prevents the learned policy deviating too much from the SFT model, and this is a popular technique to alleviate reward over-optimization (Jaques et al., 2019; Stiennon et al., 2020).

**PPO.** Proximal policy optimization (PPO) (Schulman et al., 2017) is an RL algorithm that uses a clipped version of the policy gradient for more conservative and stable learning. It becomes a standard algorithm for RL fine-tuning in RLHF that optimizes the modified (cumulative) reward

$$r_\phi(y|c) - \sum_{t=T_c+1}^{T} \beta\Big( \log \pi_\theta(y_t|c \oplus y_{1:t-1}) - \log \pi^{\text{SFT}}(y_t|c \oplus y_{1:t-1})\Big) \tag{4}$$

where the reward model gives sparse rewards and the policy constraint yields dense rewards. PPO is an on-policy algorithm where the policy gradient is estimated based on the samples collected by the current policy $\pi_\theta$.

---

**Algorithm 1** Proximal policy optimization (PPO)

---

**for** iteration $= 1, 2, \cdots$ **do**
    Fill the buffer $\mathcal{B}$ with samples collected by the current language model $\pi_\theta$
    Update $\pi_\theta$ using PPO w.r.t. the cumulative reward defined in Equation equation 4 based on $\mathcal{B}$
**end for**

---

## 4 METHODS

Our method is motivated by the observation that the reward model is more reliable for the responses assigned with high/low rewards (cf. Figure 1). Consequently, we conjecture that, if we wrap the policy with proper filtration during policy optimization of RLHF, the reward model can avoid yielding unreliable rewards and thus give better signal to guide policy learning.

**Policy filtration.** Given an unfiltered policy model $\pi_\theta(y|c)$ that generates responses $y$ to the context $c$, we denote the corresponding filtered policy as $\mu_\theta(y|c)$. We consider a family of policy filtration, from which we can sample responses to the context $c$ as follows: We first sample $N$ responses from $\pi_\theta(\cdot|c)$ and rank them by the reward model $R_\phi$, obtaining $y_1, \cdots, y_N$ with $R_\phi(y_1|c) \geq \cdots \geq R_\phi(y_N|c)$. Then, given a weight vector $\mathbf{w} = (w_1, \cdots, w_N)$ satisfying $\sum_{i\in[N]} w_i = 1$, we sample a one-hot vector $\mathbf{z} = (z_1, \cdots, z_N)$ from the categorical distribution parameterized by $\mathbf{w}$ such that $\mathbb{P}[z_i = 1] = w_i$. At last, the filtered policy $\mu_\theta(\cdot|c)$ yields the response selected by $\mathbf{z}$ following $y = \sum_{i\in[N]} z_i y_i$.

We can define several filtered policies under this family. Specifically, we obtain the best-of-$N$ (BoN), best-random (BR), and best-worst (BW) filtered policy by setting the weight vector to $\mathbf{w}^{\text{BoN}} = (1, 0, \cdots, 0)$, $\mathbf{w}^{\text{BR}} = \left(\frac{1}{2}, \frac{1}{2(N-1)}, \cdots, \frac{1}{2(N-1)}\right)$, and $\mathbf{w}^{\text{BW}} = \left(\frac{1}{2}, 0, \cdots, 0, \frac{1}{2}\right)$ respectively.

**Training objective.** Since our target is to learn a good filtered policy $\mu_\theta$, we consider the follow objective:

$$\max_{\theta} \mathbb{E}_c \mathbb{E}_{y\sim\mu_\theta(\cdot|c)} \left[r_\phi(y|c) - \beta D_{\text{KL}}(\mu_\theta(\cdot|x)||\pi^{\text{SFT}}(\cdot|x))\right]. \tag{5}$$

In practice, use the samples collected by the filtered policy $\pi_\theta$ as if they were collected by $\mu_\theta$ in the original PPO algorithm. This leads to Policy Filtration Proximal Policy Optimization (PF-PPO) listed in Algorithm 2, which is an algorithm that only modifies the sampling process of PPO.

---

**Algorithm 2** Policy Filtration Proximal policy Optimization (PF-PPO)

---

**for** iteration $= 1, 2, \cdots$ **do**
    Fill the buffer $\mathcal{B}$ with samples collected by the current language model $\mu_\theta$
    Update $\pi_\theta$ using PPO w.r.t. the cumulative reward defined in Equation equation 4 based on $\mathcal{B}$
**end for**

---

|  | No filter | BoN filter | BR filter | BW filter |
|---|---|---|---|---|
| SFT policy | 0.886 | 0.454 | 0.922 | **0.952** |
| Middle RLHF policy | 0.907 | 0.389 | 0.935 | **0.956** |
| Final RLHF policy | 0.876 | 0.431 | 0.916 | **0.946** |

Table 1: The coefficient of determination ($R^2$) of unfiltered policy $\pi_\theta$ and different filtered policies $\mu_\theta$ between the rewards given by the reward model and the actual scores. This metrics correlates well with the final performance (see Section 5) and helps us to determine the weight vector (or the policy filtering strategy) in our algorithm PF-PPO.

**Weight choice.** By defining different weight vectors $\mathbf{w}$, we can obtain different policy filtering strategies for PF-PPO. Our objective is to choose a weight vector $\mathbf{w}$ such that the accuracy of the reward model on the responses generated by the filtered policies can be maximized. To measure this accuracy, we calculate the coefficient of determination (aka R-squared or $R^2$) (Draper, 1998) between the rewards and the actual scores of the responses generated by the policy. $R^2$ measures how well the actual scores can be predicted by the rewards with a linear model. Specifically, given a set of responses $\{(c_i, y_i)\}$ sampled from the filtered policy $y_i \sim \mu_\theta(\cdot|c_i)$, we can collect the corresponding reward $R_i := R_\phi(y_i|c_i)$ and the actual score $s_i$. Then, we fit a linear model $f$ to predict the actual score based on the reward and denote the predicted score as $\hat{s}_i = f(R_i)$. The R-squared is calculated as $1 - \frac{\sum_i (s_i - \hat{s}_i)^2}{\sum_i (s_i - \bar{s})^2}$ where $\bar{s}$ is the average of actual scores. Since PF-PPO optimizes the policy based on the rewards on these responses, how well these rewards indicate the actual performance is closely related to the final performance of our algorithm. We find $R^2$ well correlates with the final performance and can imply the level of reward over-optimization of the subsequent RLHF algorithm, therefore serving as a useful metrics to determine the weight vector used in PF-PPO.

To select a weight vector, we first checkpoint three policies $\pi_\theta$ collected from different stages of a standard RLHF process and collect responses using filtered policies $\mu_\theta$ in combination with different policy filtering strategies. Then, we group the responses with similar rewards, record the average actual score and reward for each group, and calculate the $R^2$ by treating each group as a sample point. We exam how different policy filtering strategies can improve the reliability of the rewards on the responses generated by the corresponding filtered policies.

We present the results in Table 1. We observe that best-random (BR) and best-worst (BW) can improve the reliability of the given reward model on sampled responses compared with unfiltered policy. The BoN strategy does not improve the $R^2$, which indicates that learning a BoN filtered policy may not result in good performance in RL, although learning for a best-of-$N$ policy using supervised learning presents good performance (Sessa et al., 2024).

## 5 EXPERIMENTS

### 5.1 BENCHMARKS

To demonstrate the effectiveness of our method, we conduct experiments on the code generation task, which is a typical reasoning task where the quality of the responses from code LLMs can be precisely measured. Specifically, we compare different algorithms on two widely used benchmarks and a new challenging benchmark:

**HumanEval benchmark and MBPP benchmark.** HumanEval (Chen et al., 2021) and MBPP (Austin et al., 2021) are two popular benchmarks for evaluating code LLMs. HumanEval consists of 164 hand-written Python problems, each of which is validated using test cases to assess the accuracy of the code generated by a code LLM in a zero-shot setting. MBPP includes 378 test

problems, each of which includes the problem description, the standard code solution, and test cases to help us evaluate the model's ability to generate code. Both benchmarks play crucial roles These two benchmarks are widely used to evaluate the performance of large language models on code generation tasks.

**LeetCode contest benchmark.** To further evaluate the capability of the model on more challenging coding problems, we construct the LeetCode Contest benchmark. This benchmark includes competition-level problems designed for human, and therefore is more challenging since it requires human-level problem understanding and code generation skills. In this benchmark, we collect 160 problems from LeetCode weekly contests from July 2022 to January 2024. For each problem, we include 100 test cases to ensure the generated code is assessed thoroughly.

## 5.2 DATASETS AND PRE-PROCESSING

For our experiments on the HumanEval and MBPP benchmarks, we select data from the 75k Magicoder-OSS-instruct dataset (Wei et al., 2023b) and the 55k evol-codealpaca-v1 dataset (Luo et al., 2023) to construct the SFT dataset, the reward model dataset, and the PPO query dataset. Specifically, we use all the 130k training samples from Magicoder-OSS-instruct and evol-codealpaca-v1 as the SFT dataset. To train a reward model, we curate 7k prompts from these 130k samples and generate five responses using the SFT model for each prompt. Following the methodology in Pal et al. (2024), we select two responses with the maximum edit distance to create response pairs for each prompt. We use these 7k prompts with generated response pairs as the reward model dataset. For policy optimization, we curate 3k prompts from the 130k samples as the PPO query dataset.

For the LeetCode benchmark, we construct LeetCode training datasets comprising 1,000 problems collected from the LeetCode website. For SFT, we use self-generated correct answers to create the SFT dataset following the methodology in Setlur et al. (2024). For reward modeling, we generate five responses using the SFT model for each of the 400 curated prompts and selected two responses with the maximum edit distance to form the response pairs for each prompt. We use these prompts and response pairs to train the reward model. Finally, we used the full 1,000 prompts as our PPO query dataset to train the code LLM.

## 5.3 IMPLEMENTATION DETAILS

We use deepseek-6.7B (Guo et al., 2024) as our base model. In the SFT phase, we train on the SFT dataset for 5 epochs with the learning rate $1 \times 10^{-5}$, resulting in the SFT policy. In the reward model training phase, we follow Ouyang et al. (2022) and train on our reward model dataset for 1 epoch with the learning rate $1 \times 10^{-5}$. In the PPO phase, we adopt the training tricks from the blog (Shen et al., 2024). Specifically, we adopt reward normalization and advantage normalization for stable training. In addition, we set the learning rate for the policy network as $5 \times 10^{-7}$ and learning rate for the value network as $9 \times 10^{-6}$. In the PPO algorithm, we collect responses for the context in the PPO query dataset and iterate through this dataset for 5 iterations (enough for convergence) and select the best checkpoints on evaluation set as the outcome policy. For each collected context-response pair, we use it to accumulate loss and gradient for 3 times on average. We use full parameter fine-tuning in all the phases. We provide the source code for all experiments in the supplementary.

## 5.4 BASELINES

We compare different variants of PF-PPO with not only reinforcement learning algorithms but also supervised fine-tuning methods and direct policy optimization methods. We use greedy decoding during inference and pass@1 (Chen et al., 2021) as the performance metrics. For fair comparison between different baselines, we re-implement these baselines with the same code base and the same datasets. We also use the same reward model and the same SFT policy if applicable.

**Supervised fine-tuning.** Starting from deepseek-6.7B, we first fine-tune this policy on the SFT dataset. Other algorithms learn based on this SFT policy. RAFT (Dong et al., 2023) and BOND (Sessa et al., 2024) train the policy to fit the best-of-$N$ (BoN) responses or the BoN policy via different supervised learning losses. RAFT maximizes the log-probability of the BoN response, whereas BOND minimizes a combination of the forward and backward KL divergence w.r.t. the BoN policy. We set the coefficient to combine these two loss terms as $\beta_{\text{BOND}} = 1.0$. BOND is an iterative

| Family | Method | HumanEval | MBPP | LeetCode |
|---|---|---|---|---|
| Supervised Fine-Tuning | SFT | 74.2 | 70.8 | 15.2 |
| | RAFT (Dong et al., 2023) | 76.9 | 71.3 | 17.8 |
| | BOND (Sessa et al., 2024) | 80.8 | 75.2 | 30.0 |
| Direct Policy Optimization | DPO (Rafailov et al., 2024) | 78.4 | 73.7 | 23.0 |
| | IPO (Azar et al., 2024) | 78.2 | 72.9 | 23.2 |
| | KTO (Ethayarajh et al., 2024) | 77.9 | 72.5 | 22.4 |
| | Iterative-DPO (Pang et al., 2024) | 78.1 | 74.8 | 23.8 |
| Reinforcement Learning | PPO-S (Hu et al., 2024) | 78.1 | 73.8 | 25.2 |
| | PPO-M (cf. Shao et al., 2024) | 80.2 | 75.0 | 29.8 |
| | PF-PPO (BoN) | 75.8 | 71.7 | 16.8 |
| | PF-PPO (BR) | **82.9** | 75.9 | **33.0** |
| | PF-PPO (BW) | 82.4 | **76.2** | 30.4 |
| SOTA (7B models) | Magicoder (Wei et al., 2023b) | 76.8 | 75.7 | |

Table 2: **The performance of different algorithms on three benchmarks.** We compare pass@1 of PF-PPO (our algorithm) against baseline methods. For each benchmark, we select the best score across 5 epochs for each method. The highest and the second highest scores on each benchmark are highlighted in **bold** and underline respectively. All experiments are based on the same code base for fair comparison, except for the scores reported by Magicoder which is the best 7B model so far.

algorithm to fit the BoN policy based on the policy of the last iteration, and we train the policy for 4 iterations.

**Direct policy optimization.** To implement direct policy optimization methods, we use our reward model dataset as the preference dataset required in these methods. We implement DPO (Rafailov et al., 2024), IPO (Azar et al., 2024), KTO (Ethayarajh et al., 2024), and iterative DPO (Pang et al., 2024). For iterative DPO, we train the DPO model for three iterations. For each iteration, we construct the preference dataset as follows: The prompts are sampled from the reward model dataset and responses are generated by the trained DPO model from the previous iteration (if exists) or the previous SFT phase.

**Reinforcement Learning.** For standard RLHF, we use the implementation from OpenRLHF (Hu et al., 2024), which incorporates several advanced PPO training techniques and has demonstrates strong performance on various benchmarks. We denote this baseline as PPO-S. For our method PF-PPO, we implement three variants (BoN, BR, and BW) as introduced in the previous section. Since PF-PPO collects multiple responses given a prompt/context, we introduce a baseline called PPO-M (PPO with multiple responses) that uses all the $N$ responses for training without filtering. Comparing with PPO-M can help us distinguish the effect of collecting multiple responses and that of filtering collected responses. The effective difference between PPO-S and PPO-M is that the buffer $\mathcal{B}$ in PPO-M contains more samples with the same context $c$ but with different responses $y$ which may provide detailed token-level instruction by comparing the responses corresponding to the same context. PPO-M can also be regarded as integrating GRPO (Shao et al., 2024) into PPO, which has been adopted by Deepseek-V2 (Zhu et al., 2024) and Qwen2 (Yang et al., 2024). We also refer the readers to Section 5.7 for the analysis on the computational efficiency of PPO-S, PPO-M, and PF-PPO.

## 5.5 EXPERIMENT RESULTS ON THREE BENCHMARKS

We present the pass@1 results of different methods on the three benchmarks in Table 2. The experiment results show that PF-PPO (BR) and PF-PPO (BW) obtain the highest scores on these benchmarks, indicating the effectiveness of our method. Furthermore, we have the following observations:

- IPO and KTO (improved versions of DPO) do not outperform DPO when trained on properly selected datasets. This indicates that appropriate dataset construction can address the weaknesses of DPO found in previous papers, enabling DPO to achieve a performance comparable to its improved versions.

- PPO-based algorithms outperform SFT-based and DPO-based algorithms in general, demonstrating that PPO is superior to these algorithms on reasoning tasks. We speculate that the good performance of PPO may stem from the generalization ability of the reward model and the value network used in PPO, which can be used to transform trajectory-level reward modeling to token-wise advantages and thus provides more fine-grained guidance. Moreover, the gap between PPO-based algorithms and the others becomes larger on the more challenging LeetCode benchmark, which further highlights the advantage of RL on complex reasoning tasks

- BOND achieves the highest score among the baseline methods. It demonstrates that iterative best-of-$N$ (BoN) distillation is an effective alignment approach. We speculate that BOND also benefits from its ability to reduce learning on samples with unreliable rewards by selecting the best candidate from a set of $N$ samples.

- Motivated by the good performance of BOND, we implement PF-PPO (BoN) as a natural attempt to apply BoN to an RL-based algorithm. However, PF-PPO (BoN) results in poor performance. This indicates that compared with SFT methods that only need good samples, bad samples for the contrastive learning purposes are also important for RL-based methods. This explains the reason why PF-PPO (BR) and PF-PPO (BW) outperform PF-PPO (BoN).

- PF-PPO (BR) and PF-PPO (BW) outperform the others with a larger gap challenging LeetCode tasks. We find that the accuracy of the reward model decreases on this benchmark since it is more difficult for the reward model to distinguish whether one response is better than another, especially when both responses contain errors. This decreases the reliability of the reward model in the moderate reward region (cf. Figure 1). Consequently, PF-PPO (BR) and PF-PPO (BW) can improve the performance in these complex reasoning tasks by avoiding learning on unreliable rewards.

## 5.6 Choosing from Different Policy Filtering Strategies

PF-PPO modifies the sampling procedure of standard PPO by sampling $N$ responses and randomly filtering responses based on their ranks. In this part, we consider other alternatives to filter by threshold or down-weight the responses with unreliable rewards in the sampling procedure.

- **Filtering based on reward thresholds.** Given a reward model, we can filter the responses based on their rewards using specified threshold. This results in three strategies, *PPO-top* that only keeps the top samples whose rewards exceeding a certain threshold, *PPO-top-random* that keeps also keeps random samples with 50% probability, and *PPO-top-bottom* that keeps top samples and bottom samples whose rewards are below another specified threshold. These strategies can be regarded as the threshold version of PF-PPO (BoN), PF-PPO (BR) and PF-PPO (BW) respectively. The thresholds are tuned coarsely to achieve good results on a separate validation set.

- **Filtering based on reward reweighting.** Compared with the above strategies that use thresholds, we consider a softer version that adjusts the sample weights based on their rewards, aiming at down-weight the samples with moderate and possibly unreliable rewards. Specifically, we increase the sample weight of the responses with rewards in the reliable region and decrease the sample weight otherwise. To achieve this goal, given a reward model $R_\phi$ that returns rewards in the range $[-1, 1]$, we assign the weight for the sample $(c, y)$ proportional to $|R_\phi(y|c)|^k$ and collect samples with these weights from the buffer $\mathcal{B}$ to train the policy network and the value network. We denote these strategies as *PPO-pow-k*.

A question then arises: how to choose a policy filtering strategy from these strategies? To answer this question, we propose to calculate the $R^2$ between the rewards and the actual scores on the samples collected by different strategies, and then choose a strategy with good results on this metrics. We can use the SFT policy as the unfiltered policy and calculate $R^2$ as described in Section 4. Since the SFT policy is obtained prior to the PPO training phase, this metric can be used to predict the results of different filtering strategies before actually conduct costly PPO training.

We compare theses strategies on HumanEval and present the performance of different policy filtering strategies and their corresponding $R^2$ in Table 3. We make the following observations: First, the $R^2$ of different strategies positively correlate with their performance in general, indicating $R^2$ can serve

| Policy filtering strategies | pass@1 on HumanEval | pass@1 on MBPP | $R^2$ based on SFT policy |
|---|---|---|---|
| PPO | 78.1 | 73.8 | 0.782 |
| PPO-M | 80.8 | 75.0 | 0.886 |
| PF-PPO (BoN) | 75.8 | 71.7 | 0.454 |
| PF-PPO (BR) | 82.9 | 75.9 | 0.841 |
| PF-PPO (BW) | 82.4 | 76.2 | 0.952 |
| PPO-top | 80.5 | 71.2 | 0.621 |
| PPO-top-random | 81.9 | 75.3 | 0.889 |
| PPO-top-bottom | 81.7 | 75.4 | 0.927 |
| PPO-pow-1 | 81.0 | 74.2 | 0.926 |
| PPO-pow-2 | 81.3 | 75.4 | 0.939 |
| PPO-pow-3 | 81.9 | 76.5 | 0.946 |

Table 3: The comparison on the pass@1 results of different policy filtering strategies on HumanEval and their corresponding $R^2$ based on the SFT policy. The background are colored based on their values with blue and red indicating the minimum and the maximum respectively.

as a tool to predict the performance of different policy filtering strategies. Second, different policy filtering strategies (except for BoN versions) improve the performance of the base PPO algorithms. This indicates that filtering samples with unreliable rewards can increase the signal-to-noise ratio of the reward model feedback and thus improve the performance. Third, PF-PPO strategies (which are rank-based) outperforms other strategies (which are threshold-based or reweighting-based). This may due to the fact that rank-based strategies are more robust to the reward distribution of the given reward model.

**Discussion.** The performance of different policy filtering strategies may vary across different tasks, different reward models, and different base models. Therefore, although we find that PF-PPO (BR) and PF-PPO (BW) are the best strategies in our setting, other policy filtering strategies may be a better choice in other settings. Therefore, a more practical procedure should be first calculate the $R^2$ using the given reward model and the corresponding SFT policy on the specific task and select candidate policy filtering strategies. Note that $R^2$ is not a perfect tool to select policy filtering strategies and we leave seeking for better predictive metrics as a future research direction.

### 5.7 FURTHER ANALYSIS

**The training process of PPO-S, PPO-M, and PF-PPO.** To provide a comprehensive view of the three algorithms, we show the training process.

We first present the training curves of PPO-S, PPO-M, and PF-PPO in Figure 2 (left). The training reward are evaluated on the samples collected by the filtered policy $\mu_\theta$ and the evaluation rewards are calculated on the unfiltered policy $\pi_\theta$. We observe that both the training reward and evaluation reward of PPO-M and PF-PPO surpass those of PPO-S. This indicates that sampling multiple responses from a context enhances the performance of the RLHF method, consistent with the findings in Shao et al. (2024). Moreover, in terms of optimizing reward for the given reward model, FP-PPO achieves a higher or equal reward compared with PPO-S and PPO-M, which indicates that the approximation made in the FP-PPO (i.e., optimizing $\pi_\theta$ as if it were $\mu_\theta$) does not induce negative effect on its capability to optimize the reward.

We also show the pass@1 results of different algorithms in Figure 2 (right). We observe that, while PF-PPO achieves a similar reward to that of PPO-M, the pass@1 result of PF-PPO exceeds that of PPO-M significantly. This results from the fact that PF-PPO optimizes on the reliable region of the reward model and thus alleviate the reward over-optimization issue.

**Computational efficiency of PPO-S, PPO-M, and PF-PPO.** PPO-S, PPO-M, and PF-PPO all collect different number of responses per query and train using different number of samples. For clarity, we list the computational complexity of these algorithms in Table 4. Note that, for all algorithms, we select the best checkpoint on the evaluation set and report the performance of this checkpoint. Combining the results in Table 4 and Figure 2, we can draw the following conclusions: First, the total computational complexity of PPO-S and PPO-M is almost the same, and the only difference is

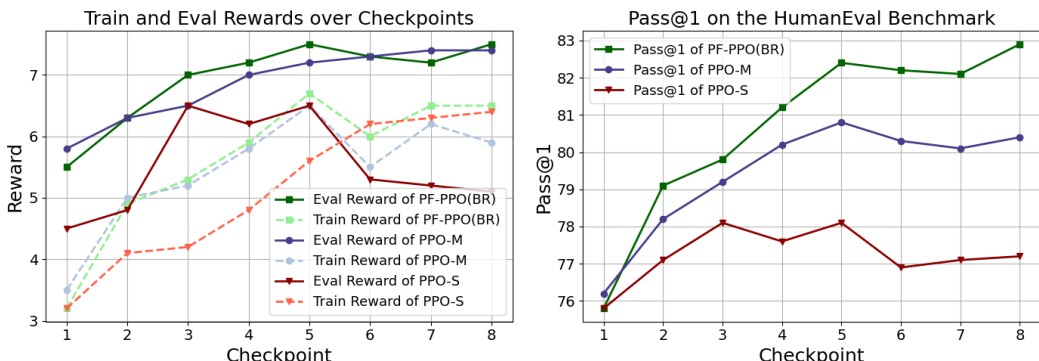

Figure 2: Left: The training and evaluation reward of PPO-S, PPO-M, and FP-PPO on HumanEval. The training reward and the evaluation reward are evaluated on the samples generated by the filtered policy $\mu_\theta$ and the unfiltered policy $\pi_\theta$ respectively. Right: The pass@1 of PPO-S, PPO-M, and PF-PPO on the HumanEval benchmark.

| | PPO-S | PPO-M | PF-PPO (BR / BW) |
|---|---|---|---|
| Queries sampled per iteration | $5n$ | $n$ | $n$ |
| Responses sampled per query | 1 | 5 | 5 |
| #Query-response pairs per iteration | $5n$ | $5n$ | $5n$ |
| Reward model forward pass per iteration | $5n$ | $5n$ | $5n$ |
| Critic forward&backward pass per iteration | $5nm$ | $5nm$ | $2nm$ |
| Policy forward&backward pass per iteration | $5nm$ | $5nm$ | $2nm$ |
| HumanEval | 100% | +2.69% | +6.15% / +5.51% |
| MBPP | 100% | +1.63% | +2.85% / +3.25% |
| LeetCode | 100% | +18.25% | +30.95% / +20.63% |

Table 4: Comparison of computational complexity and the performance of PPO-S, PPO-M, and PF-PPO. We use $n$ to denote the number of queries in the PPO query dataset, and use $m$ to denote the number of PPO epochs (i.e., each query-response pair is used to accumulate loss and gradient for $m$ times on average). PPO-M and PF-PPO collect $N = 5$ responses per query, and PF-PPO select 2 out of the $N = 5$ responses (on average) for network update. We also show the performance improvement of PPO-M and PF-PPO based on PPO-S.

that PPO-M is more likely to learn from different responses with the same query in the same batch or adjacent batches, which improves the performance. Second, the computational complexity of PF-PPO is less than that of PPO-S and PPO-M, while PF-PPO outperforms them. This indicates the effectiveness of our method.

# 6 CONCLUSION

In this paper, we propose a new reinforcement learning with human feedback (RLHF) method, **Policy Filtration for Proximal Policy Optimization (PF-PPO)**, aimed at mitigating the adverse effects of reward noise. When training the reward model using the Bradley-Terry approach, the reward signal is generally more reliable in the high or low reward regions but less reliable in the moderate reward regions. Motivated by this observation, we adopt a rank-based method to selectively use sample from these reliable regions more in PPO to improve the quality of the signal provided by the reward model. We conduct comprehensive experiments on code generation tasks, demonstrating that PF-PPO outperforms existing baselines. Additionally, we analyze PF-PPO, standard PPO, and PPO with multiple responses in details and show that filtering samples with unreliable rewards can improve the performance of the outcome policy.

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

## A  REWARD MODEL

The design of our algorithm is motivated by the observation that the reward model is less reliable when it yields moderate rewards. To provide more evidence that this property is universal across a broader range of benchmarks, we provide analyze the reward function on the MBPP and LeetCode benchmarks. We repeat the process in Figure 1 on MBPP and LeetCode and plot the figures in Figure 3. Note that we train different reward functions based on the datasets from these two benchmarks. We observe that the property holds on these two additional benchmarks, indicating this property may extend to broader fields.

Intuitively, this property should be universal to a broader range of tasks. For code generation tasks, it is quite common that some samples (e.g., the response matches the known correct answer or the response contains an obvious error) are easier to evaluate than others (e.g., the response tries to solve the problem by a novel approach). Therefore, those samples that are hard to evaluate by human should also be hard instances for the reward model.

## B  EXPERIMENT RESULTS ON MATH REASONING TASKS

To evaluate the effectiveness of PF-PPO in other domains, we applied PF-PPO to solve math problems. We use Qwen1.5-7B (Team, 2024) as the SFT model and Ape210K (Zhao et al., 2020) and CMATH (Wei et al., 2023a) as the evaluation benchmarks. Other experimental settings are the same as Zhou et al. (2024). We use three types of reward models: the original reward model (ORM) that is trained on preference datasets using a Bradley–Terry model (Bradley & Terry, 1952), an oracle model (Oracle) that extracts the final answer from the response and compares it with the ground truth, and a combined reward model (CRM) that integrates the above two models, similar to the approach used in Qwen-Math (Yang et al., 2024). We compare PF-PPO to the standard PPO (PPO-S) using these reward models. We select the policy filtration strategy according to the procedure described in our main text, and choose the BR variant of PF-PPO.

|                  | Ape210K | CMATH |
|------------------|---------|-------|
| PPO-S + ORM      | 84.1    | 92.3  |
| PF-PPO + ORM     | **86.2**| **95.1**|
| PPO-S + Oracle   | 82.1    | 90.8  |
| PF-PPO + Oracle  | **83.8**| **91.2**|
| PPO-S + CRM      | 83.9    | 93.1  |
| PF-PPO + CRM     | **84.3**| **94.2**|

Table 5: Comparison between PF-PPO and PPO-S on two math benchmarks (Ape210K and CMATH) using three different reward functions (the original reward model, the oracle model, and the combined reward model). Better results for each reward model is highlighted in **bold**.

We can observe that PF-PPO consistently outperforms the PPO algorithm on these two benchmarks across different reward models. In addition, the experiment results indicate that even if we can have access to the ground truth, using the oracle as the reward function does not perform as well as using a reward model (either the original reward model or the combined model). This finding is consistent with experiment results in Qwen-Math (Yang et al., 2024) and Deepseek-Math (Shao et al., 2024).

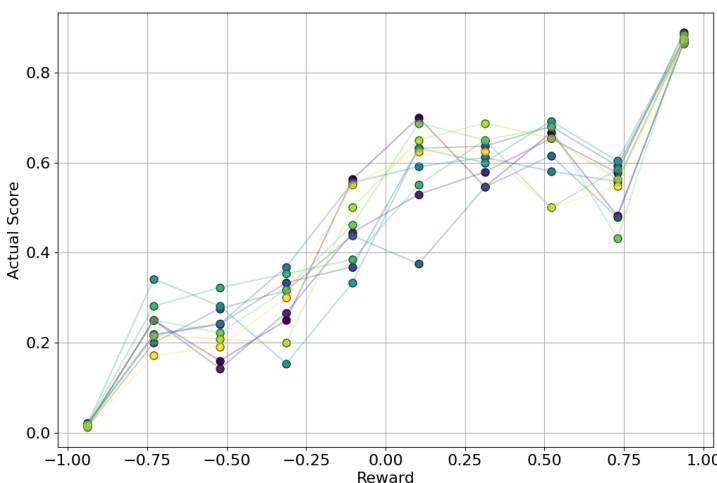

(a) The actual scores vs. the reward values for the reward model evaluated on MBPP

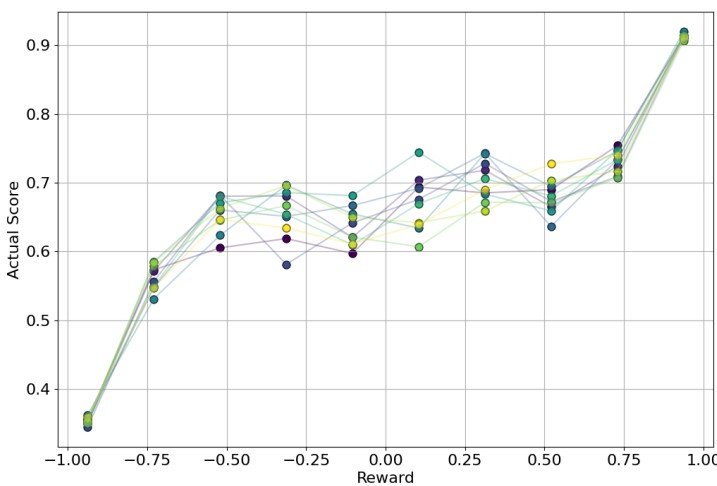

(b) The actual scores vs. the reward values for the reward model evaluated on LeetCode

Figure 3: We provide additional evidence that the reward model is less reliable when it yields moderate rewards than when it yields high or low rewards. We conduct the same statistics as in Figure 1 but on different benchmarks. Specifically, the reward models for the MBPP and LeetCode benchmarks are trained separately using the corresponding datasets for these two benchmarks. The MBPP and LeetCode benchmarks contains 378 and 1570 prompts respectively and we collect 10 responses for each prompt using a fine-tuned policy. We group the responses with similar rewards and calculate the average of their actual scores (i.e., the average correctness), indicating each group by one point. To evaluate the reliability of the reward model, we repeat the process ten times resulting in ten lines.

## C  QUALITATIVE RESULTS

In this section, we provide qualitative results on 1) how responses with high/middle/low rewards look like and why responses with middle rewards are unreliable; and 2) the qualitative difference between the code generated by the PF-PPO policy and the standard PPO (PPO-S) policy.

### C.1  ANALYSIS ON THE THE RESPONSES ASSOCIATED WITH DIFFERENT REWARDS

We present a prompt along with several responses, including a correct response but assigned with a low reward, an incorrect response but assigned with a high reward, an incorrect response with a low reward, and a correct response with a high reward. The prompt describes a coding problem that requires to convert fractions to decimals.

We have the following findings:

- For the correct response but assigned with a low reward, the generated code is less clear and harder to read. For example, the code may mix several steps into one line.

- For the incorrect response but assigned with a high reward, the response incorrectly mixes two correct approaches. This mistake can hardly be identified by the reward model (and even GPT-4).

- For the incorrect response assigned with a low reward, the response contains an obvious mistake which is easily detected by the reward model.

We also provide detailed analysis into the solutions to this problem. The given prompt is a coding task to convert fraction to decimal.

```
1  Given two integers representing the numerator and denominator of a
       fraction, return the fraction in string format.
2  If the fractional part is repeating, enclose the repeating part in
       parentheses.
3  If multiple answers are possible, return any of them.
4  It is guaranteed that the length of the answer string is less than 104
       for all the given inputs.
5
6  Please complete the following python code precisely:
7  ```python
8  class Solution:
9      def fractionToDecimal(self, numerator: int, denominator: int) -> str:
10 ```
```

This is a correct response with a high reward. This solution is thorough with clear variable names and properly structured steps. It is easier to read due to breaking down steps explicitly such as calculating the integer part and handling the remainder.

```
1
2
3  To solve the problem of converting a fraction to its decimal
       representation, we can follow these steps:
4
5  1. **Determine the sign of the resultant decimal**: If both the numerator
       and the denominator are negative, the result should be positive.
       Otherwise, the result will have a leading minus sign if and only if
       either the numerator or the denominator is negative.
6
7  2. **Convert the numerator and denominator to positive values**: Since
       the division process involves integer division and the decimal part
       requires modulus operation, both numerator and denominator should be
       made positive for easier computation.
8
9  3. **Calculate the integer part**: The integer part is the result of
       integer division of the numerator by the denominator.
10
```

4. **Manage the remainder**: As we perform integer division, the
   remainder will be used to form the fractional part. This remainder
   will be treated as the new numerator for the next step.

5. **Use a dictionary to manage repetition**: To detect recurring
   decimals, we need to keep track of all remainders we have seen and
   their respective positions in the decimal expansion. If a remainder
   repeats, it indicates a recurring decimal.

6. **Build the result string**: Start with the integer part, then append
   a decimal point. Then, use a loop to compute each digit of the
   fractional part. Use the dictionary to detect and mark any recurring
   decimals.

7. **Return the result**: Finally, return the constructed string for the
   result.

Below is the complete Python code for the `fractionToDecimal` method:

```python
class Solution:
    def fractionToDecimal(self, numerator: int, denominator: int) -> str:
        if numerator == 0:
            return "0"

        # Determine the sign of the result
        result_sign = "-" if (numerator < 0) ^ (denominator < 0) else ""

        # Make both numerator and denominator positive
        numerator = abs(numerator)
        denominator = abs(denominator)

        # Calculate the integer part of the result
        integer_part = numerator // denominator
        remainder = numerator % denominator

        # If there's no remainder, return the integer part
        if remainder == 0:
            return result_sign + str(integer_part)

        # Prepare to build the fractional part
        result = []
        result.append(".")  # Start with the decimal point
        remainder_map = {}  # To keep track of remainders and their
            positions

        # Process each place value in the fractional part
        position = 0
        while remainder not in remainder_map:
            remainder_map[remainder] = position
            remainder *= 10
            result_digit = remainder // denominator
            result.append(str(result_digit))
            remainder %= denominator
            if remainder == 0:
                break
            position += 1

        # If there's a repeat, insert parentheses
        if remainder in remainder_map:
            repeat_start = remainder_map[remainder]
            result.insert(repeat_start + 1, "(")
            result.append(")")

        return result_sign + str(integer_part) + "".join(result)
```

```
66  ```
67
68  This code correctly handles edge cases like zero denominator, negative
        numbers, repeating decimals, and ensures that the result is less than
        10^4 characters long.
```

This is a correct response but assigned with a low reward. Compared with the previous response with high reward, this response mixes multiple operations in one line, making it harder to understand (e.g., Line 34).

```python
1   To solve the problem of converting a fraction to its decimal
        representation, we can use the following approach:
2
3   1. Handle the sign of the result.
4   2. Calculate the integral part of the result, which is the part before
        the decimal point.
5   3. Handle the case where the fractional part is repeating.
6   4. Calculate the fractional part and handle cycles.
7
8   Here's the completed Python code:
9
10  ```python
11  class Solution:
12      def fractionToDecimal(self, numerator: int, denominator: int) -> str:
13          # Handle sign of result
14          sign = '-' if numerator * denominator < 0 else ''
15          numerator, denominator = abs(numerator), abs(denominator)
16
17          # Calculate integral part
18          integral_part = numerator // denominator
19
20          # Calculate fractional part
21          remainder = numerator % denominator
22          if remainder == 0:
23              return sign + str(integral_part)
24
25          # Use a dictionary to track remainders and their positions
26          # in the fractional part to detect cycles
27          seen_remainders = {}
28          result = []
29          index = 0
30
31          while remainder not in seen_remainders:
32              seen_remainders[remainder] = index
33              result.append(str(remainder * 10 // denominator))
34              remainder = (remainder * 10) % denominator
35              if remainder == 0:
36                  break
37              index += 1
38
39          # If there's a cycle, insert the parentheses
40          if remainder in seen_remainders:
41              index = seen_remainders[remainder]
42              result.insert(index, '(')
43              result.append(')')
44
45          return sign + str(integral_part) + '.' + ''.join(result)
46  ```
47
48  ### Explanation:
49
50  - We first handle the signs of the numerator and denominator.
51  - We then calculate the integral part of the result by integer division.
```

```
52  - If the remainder is zero, the fraction is an integer and we return it
        as a string.
53  - We use a dictionary `seen_remainders` to keep track of the positions
        where each remainder appears, allowing us to detect cycles.
54  - If a cycle is detected, we insert parentheses at the start of the
        repeating part.
55  - Finally, we assemble the result as a string and return it.
```

This is an incorrect response but assigned with a high reward. In Line 32, the decimal point is added to the `result` list but is not later counted when getting wrapped by the parentheses, leading to the wrong format. This is a mixture of two correct approaches, one that adds the decimal points to `result` but sets an offset for this (cf. Line 44 and Line 62 in the first response) and one that outputs the decimal point separately (cf. Line 45 in the second response).

```
1   To solve the problem of converting a fraction to a decimal string, we
        need to handle the following cases:
2
3   1. **Integers**: If the numerator and denominator are both integers, the
        decimal representation is straightforward: divide the numerator by
        the denominator and convert the result to a string.
4
5   2. **Decimal Representation**: If the division produces a remainder, it
        indicates a repeating decimal pattern. Use a dictionary to keep track
        of the remainders and their corresponding positions in the resultant
        decimal string. Whenever a remainder repeats, enclose the repeating
        part in parentheses.
6
7   3. **Handling Negative Numbers**: Handle both positive and negative
        numbers appropriately.
8
9   Here is the complete code for the solution:
10
11  ```python
12  class Solution:
13      def fractionToDecimal(self, numerator: int, denominator: int) -> str:
14          # Handle sign of the result
15          sign = '-' if numerator * denominator < 0 else ''
16          numerator, denominator = abs(numerator), abs(denominator)
17
18          # Calculate integral part
19          integral_part = numerator // denominator
20          numerator %= denominator
21
22          # Handle zero remainder
23          if numerator == 0:
24              return sign + str(integral_part)
25
26          # Prepare list to store the decimal digits
27          result = []
28          # Dictionary to keep track of remainders and their positions
29          remainder_map = {}
30
31          # Initialize the decimal part
32          result.append('.')
33          position = 0
34
35          # Loop until the remainder repeats or becomes zero
36          while numerator > 0:
37              # If the remainder has already appeared before, it's a
                    repeating part
38              if numerator in remainder_map:
39                  result.insert(remainder_map[numerator], '(')
40                  result.append(')')
41                  break
```

```
42              # Record the position of the current remainder
43              remainder_map[numerator] = position
44              numerator *= 10
45              result.append(str(numerator // denominator))
46              numerator %= denominator
47              position += 1
48
49
50          return sign + str(integral_part) + ''.join(result)
51  ```
52
53  ### Explanation:
54  - We use a dictionary 'remainder_map' to track the position of each
        remainder seen. This helps us identify the repeating part of the
        decimal if it exists.
55  - The loop continues until the remainder becomes zero or repeats itself.
56  - If the remainder repeats, we insert the opening '(' immediately before
        the repeated part, and close it with ')'.
57  - The result is assembled from the integral part, the decimal point, and
        the decimal digits or parenthesized pattern.
```

This is an incorrect response with a low reward. In Line 59-61, the program contains an obvious error that it cannot handle the scenario where there exists a decimal part but does not contain any repeating part.

```
1
2
3   To solve the problem of converting a fraction to its decimal
        representation, we need to consider the following steps:
4
5   1. **Handling Zero Denominator**: If the denominator is zero, it's
        impossible to divide, so we should return "NaN" (Not a Number) or
        handle it according to the problem constraints.
6
7   2. **Handling Negative Numbers**: If either the numerator or denominator
        is negative, the sign of the result should be negative.
8
9   3. **Calculate the Integral Part**: The integral part is the part before
        the decimal point, which can be calculated by integer division.
10
11  4. **Calculate the Fractional Part**:
12      - Use a dictionary to keep track of the positions where the same
            remainder started repeating.
13      - Divide the remainder by the denominator repeatedly, building the
            fractional part digit by digit.
14      - If the remainder repeats, it indicates a recurring decimal, and we
            should insert parentheses to mark it.
15
16  Here's the complete code implementation:
17
18  ```python
19  class Solution:
20      def fractionToDecimal(self, numerator: int, denominator: int) -> str:
21          # Handle zero denominator
22          if denominator == 0:
23              return "NaN"
24
25          # Handle signs
26          result_sign = '-' if (numerator < 0) ^ (denominator < 0) else ''
27          numerator, denominator = abs(numerator), abs(denominator)
28
29          # Calculate the integral part
30          integral_part = numerator // denominator
31          numerator %= denominator
```

```
        # Initialize the fractional part list and a dictionary to track
            remainders
        fractional_part = []
        remainder_index = {}

        # Continue while there is a remainder
        position = 0
        while numerator > 0:
            # If the remainder has already occurred before, it's a
                repeating decimal
            if numerator in remainder_index:
                break

            # Record this remainder's position
            remainder_index[numerator] = position

            # Calculate the next digit
            numerator *= 10
            fractional_part.append(str(numerator // denominator))
            numerator %= denominator
            position += 1

        # Build the result
        if not fractional_part:
            # No repeating part, return integral part
            return result_sign + str(integral_part)
        else:
            # Repeating part detected
            repeating_start = remainder_index[numerator]
            return result_sign + str(integral_part) + '.' + ''.join(
                fractional_part[:repeating_start]) + \
                    '(' + ''.join(fractional_part[repeating_start:]) + ')'
```

### Explanation:

- **Integral Part**: We calculate the integral part as `numerator //
  denominator`.
- **Non-repeating Part**: We use a dictionary to track positions of
  remainders, and if the same remainder appears again, we know the
  fraction is repeating.
- **Repeating Part**: When a repeating part is detected, we insert
  parentheses around the repeating part.
- **Edge Cases**: We handle edge cases such as negative numbers, zero
  denominator, and ensuring the result is within the constraints of the
  problem.

This implementation correctly handles both whole number and repeating
  decimal results, adhering to the problem statement's requirements.

## C.2 ANALYSIS ON THE RESPONSES GENERATED BY PF-PPO COMPARED WITH STANDARD PPO (PPO-S)

We compare the answers from PF-PPO and PPO-S respectively for the same prompts, and conclude
that the answer from the PF-PPO policy is more inclined to follow a standard approach and the
response is more concise (or shorter), making it easier to understand and implement.

Specifically, we present their responses for the prompt that request the agent to write a code to *find
all safe nodes in a directed graph*. The two responses given by PF-PPO and PPO-S are both correct.
However, the difference is that PF-PPO adopts the deep first search (DFS) method while PPO-S
adopts the topological sorting approach. The logic is simpler for DFS and the implementation is

easier, making it easier to understand. Both approaches have roughly equivalent time and space costs, but the DFS method is slightly more space efficient.

These findings suggest an advantage in terms of readability and implementation simplicity when using the PF-PPO policy, particularly for those familiar with recursive techniques.

Prompt (find all safe nodes in a directed graph):

```
There is a directed graph of n nodes with each node labeled from 0 to n -
    1. The graph is represented by a 0-indexed 2D integer array graph
    where graph[i] is an integer array of nodes adjacent to node i,
    meaning there is an edge from node i to each node in graph[i].
A node is a terminal node if there are no outgoing edges. A node is a
    safe node if every possible path starting from that node leads to a
    terminal node (or another safe node).
Return an array containing all the safe nodes of the graph. The answer
    should be sorted in ascending order.

Please complete the following python code precisely:
```python
class Solution:
    def eventualSafeNodes(self, graph: List[List[int]]) -> List[int]:
```
```

The response from PF-PPO:

```
To solve the problem of finding all safe nodes in a directed graph, we
    can use a topological sorting approach. The idea is to explore the
    graph using depth-first search (DFS) and maintain a status for each
    node indicating whether it is safe or not. Here's the step-by-step
    breakdown of the solution:

1. **Initialize Status**: We will use a status array to track the safety
    of each node. 0 means the node is not visited yet, 1 means it is a
    safe node, and -1 means it is currently in the recursion stack (i.e.,
     being processed in the current DFS path).

2. **DFS Function**:
   - For each node, if it's safe or not is already determined (i.e.,
       status is not 0), return the status.
   - Set the status to -1 to indicate that we are currently processing it
       .
   - For each neighbor of the current node, call DFS. If any neighbor
       returns false, it indicates that this path has a cycle and the
       node is not safe.
   - If no cycle is found for all neighbors, mark the node as safe and
       return true.

3. **Iterate over all nodes**: Starting from each node that is not
    visited, perform a DFS.

4. **Collect Results**: All nodes that return true from DFS are safe
    nodes. Collect these node indices and return them in ascending order.

Here's the complete Python code for the solution:

```python
from typing import List

class Solution:
    def eventualSafeNodes(self, graph: List[List[int]]) -> List[int]:
        n = len(graph)
        # status: 0 for unvisited, 1 for safe, -1 for processing
```

```
        status = [0] * n

        def dfs(node):
            # If the node is already processed (i.e., marked as safe or
                unsafe), return its status
            if status[node]!= 0:
                return status[node] == 1

            # Mark the node as being processed
            status[node] = -1

            # Visit all neighbors
            for neighbor in graph[node]:
                # If any neighbor is unsafe or in a cycle, the node is
                    unsafe
                if not dfs(neighbor):
                    return False

            # All neighbors are safe, mark the node as safe and return
                true
            status[node] = 1
            return True

        # Try to DFS from each node
        safe_nodes = []
        for i in range(n):
            if dfs(i):
                safe_nodes.append(i)

        return safe_nodes
```

### Explanation:
- **Topological Sorting**: We are using DFS to check if a path exists
  from a node to a terminal node. If a cycle is detected, the node is
  not safe.
- **DFS Function**: The `dfs` function determines if a node is safe by
  checking all of its neighbors recursively. If any neighbor is unsafe
  or in a cycle, the node is unsafe.
- **Status Tracking**: The status array is used to track if a node is
  safe or not, and helps avoid redundant computations.
- **Result Collection**: All nodes that return `True` from the `dfs`
  function are safe nodes, and are added to the result list.

This approach efficiently identifies safe nodes by using a topological
  sorting approach and a depth-first search approach to detect cycles.

The response from PPO-S:

To solve this problem, we can use a \red{topological sort approach}
  combined with a depth-first search (DFS) approach to determine the
  terminal and safe nodes in the graph. The topological sort ensures
  that we process nodes in the order of their out-degree, which is
  suitable for directed acyclic graphs (DAGs), where no cycles can form
  .

The idea is to traverse the graph using DFS, keeping track of the current
   path during traversal to detect cycles. Nodes that do not lead to a
  cycle are considered terminal, and nodes that are part of a cycle or
  are part of a path that leads to a cycle are considered unsafe.

Here's the Python implementation of the solution:

```python
from typing import List
from collections import defaultdict, deque

class Solution:
    def eventualSafeNodes(self, graph: List[List[int]]) -> List[int]:
        # Initialize a dictionary to store the out-degree of each node
        out_degree = defaultdict(int)
        # Initialize a dictionary to store the dependencies (incoming
            edges) of each node
        dependencies = defaultdict(list)

        n = len(graph)
        # Build the dependencies and out-degree dictionaries
        for u in range(n):
            out_degree[u] = len(graph[u])
            for v in graph[u]:
                dependencies[v].append(u)

        # Initialize a queue for BFS and a set to keep track of safe
            nodes
        queue = deque()
        safe_nodes = set()

        # Add all nodes with no outgoing edges (terminals) to the queue
        for u in range(n):
            if out_degree[u] == 0:
                queue.append(u)

        # BFS to find safe nodes
        while queue:
            u = queue.popleft()
            safe_nodes.add(u)
            # For each node v that depends on u, decrement its out-degree
            # If v's out-degree becomes zero, it becomes a terminal
            for v in dependencies[u]:
                out_degree[v] -= 1
                if out_degree[v] == 0:
                    queue.append(v)

        # Sort and return safe nodes
        return sorted(safe_nodes)
```

### Explanation:
1. **Dependencies**: We build a `dependencies` dictionary, where each key is a node and the values are a list of nodes that depend on the key. This is used to track back edges.
2. **Out-degree**: We also keep track of the out-degree of each node. This is used to determine which nodes are terminal (i.e., have no outgoing edges).
3. **Topological Sort (BFS)**: We start by adding all nodes with no outgoing edges (terminals) to the queue. We then process each node from this queue. For each node, we mark it as safe, reduce the out-degree of its dependencies, and if any of those dependencies become terminals (i.e., have no outgoing edges), they are added to the queue. This ensures we process all nodes that are part of cycles and prevent future revisits to nodes involved in such cycles.
4. **Result**: The safe nodes are collected and sorted and returned as the final result.

This approach ensures that we correctly identify terminal nodes and use them to establish safe nodes, as well as handle cycles in the graph.

