# OpenReview forum: "Policy Filtration in RLHF to Fine-Tune LLM for Code Generation"
_ICLR.cc/2025/Conference — Submitted to ICLR 2025_

### Official Review · Reviewer_FPLk · 2024-11-03

**Soundness:** 1
**Presentation:** 1
**Contribution:** 2
**Rating:** 3
**Confidence:** 4

**Summary:**

The paper proposed to filter reliable RLHF samples to mitigate inaccurate reward model predictions

**Strengths:**

The idea of dropping samples with unreliable reward predictions during RLHF is novel.

**Weaknesses:**

1. [Soundness] The proposed method was largely based on an observed property: "the reward model is more reliable for the responses assigned with high/low rewards". However, this argument is not well-defined, and the property works in what scope is not discussed. Specifically,
  - What do high/low rewards exactly refer to, esp. across different benchmarks and tasks?
  - Is this property universal for different reward models across different tasks? The paper do not clearly state when this property is applicable, and there is no sufficient quantifiable analysis on this property. Therefore the universality of this property remains unclear, and I question about it. In particular, does this property persists across benchmarks and tasks? The proposed method appears general, but is only evaluated on code generation, what about other tasks?
  - I don't understand how Figure 1 supports the suggested property. I notice Figure 1 show the comparisons between reward estimations and labels, but involve very small number of prompts (164) in only one benchmark (HumanEval). The plots are noisy, and the paper do not clearly explain how such plot support the suggested property.

2. [Contribution] The paper proposed three heuristic-based designs of policy filtration.
- The design is ad-hoc, and the paper lacks clear explanations on why specific design work better or worse.
- Ad-hoc designs limit the contribution of the proposed method. Specifically, if I focus on performance on other aspects, e.g., math reasoning, helpfulness, how to design the policy filtration weight vectors? In the current state, the contribution is limited. I think at least make more in-depth analysis on why specific designs work, and/or discover essential principles that make policy filtration work well, so that it informs how the specific policy filtration can be designed, across tasks.

3. [Presentation] The foundation of the proposed method is the reliability measure (or accuracy) for reward model prediction. The paper propose to use the the coefficient of determination (R-squared), but the paper do not include details for computing this measure, and do not clearly state the motivation and/or why this measure is valid to measure the reliability of reward estimate. For instance, why choose this measure? why not other measures, e.g., errors of the statistical estimation? Lack of clarity hinders understanding for the audience who are not familiar with R-squared. There are many other presentation issues, which makes the paper hard to understand.

**Questions:**

Please see the questions above.

---

> ### Author Response · Authors · 2024-11-22
> **Author's Response to Reviewer FPLk (1/2)**
>
> We would like to thank the reviewer for valuable feedback and for the comments and questions for additional clarity. Below, we address the key comments and questions.
>
> > What do high/low rewards exactly refer to, esp. across different benchmarks and tasks?
>
> We subtract the reward by its mean and pass it through tanh activation, resulting in the reward range  [-1, +1]. The high/low rewards refers to the high/low values within this range. In addition, it is a standard process to normalize the reward model before using it in the RLHF process (cf. the text under Table 3 of InstructGPT [1]). Demean and clip is also a general practice for RL training (see also [2,3]). As long as the reward model in other tasks or benchmarks follows this preprocessing convention, the high/low rewards are well defined.
>
>
> > Is this property universal for different reward models across different tasks? The paper do not clearly state when this property is applicable, and there is no sufficient quantifiable analysis on this property. Therefore the universality of this property remains unclear, and I question about it. In particular, does this property persists across benchmarks and tasks? The proposed method appears general, but is only evaluated on code generation, what about other tasks?
>
> This property (that the reward model is less reliable when it gives middle rewards) is associated with the fact that some samples are easier to evaluate (e.g., the response corresponds to the known standard solution or the response contains obvious errors) than others (e.g., the response tries to solve the problem using a novel approach). See Appendix C1 in the revised paper for details.
> To further validate this property on other benchmarks, we draw similar plots for the MBPP and LeetCode benchmarks in Appendix A in our revised paper and observe that this property holds. Further, we provide additional experiment results on math reasoning tasks in Appendix B and the success also indicates the universality of our method.
>
> > I don't understand how Figure 1 supports the suggested property. I notice Figure 1 show the comparisons between reward estimations and labels, but involve very small number of prompts (164) in only one benchmark (HumanEval). The plots are noisy, and the paper do not clearly explain how such plot support the suggested property.
>
> Figure 1 illustrates that, when the reward model gives a high (>0.8) or low (<-0.8) rewards, this reward is trustable since it well aligns with the actual score; otherwise, the reward model is less reliable. Actually, Figure 1 is a summary based on 1640 samples since we generate 10 responses for each prompt. This makes the result statistically significant. In addition, it is not trivial to scale up the number of prompts/problems for code generation benchmarks since the benchmark needs to contain a code interpreter, test cases, and correct answers.
>
> > The design is ad-hoc, and the paper lacks clear explanations on why specific design work better or worse.
>
> Actually, our natural choice is to keep the samples with the highest and lowest rewards (i.e., the BW version of our algorithm) based on the observation that the reward model is not reliable when it gives moderate scores to some samples. To validate our conjecture, we conduct experiments on a set of more designs. The experiment result verifies our conjecture. We will provide more explanation on the comparison between different designs.

---

> ### Author Response · Authors · 2024-11-22
> **Author's Response to Reviewer FPLk (2/2)**
>
> > Ad-hoc designs limit the contribution of the proposed method. Specifically, if I focus on performance on other aspects, e.g., math reasoning, helpfulness, how to design the policy filtration weight vectors? In the current state, the contribution is limited. I think at least make more in-depth analysis on why specific designs work, and/or discover essential principles that make policy filtration work well, so that it informs how the specific policy filtration can be designed, across tasks.
>
> Following the reviewer’s suggestion, we conduct additional experiments on math reasoning tasks and we update the experiment results to Appendix B. Specifically, we use Qwen1.5-7B [6] as the SFT model and Ape210K [7] and CMATH [8] as the evaluation benchmarks. Other training settings (including the training datasets) are the same as [9]. We also try different reward functions (see details in our revised paper). We observe that PF-PPO consistently outperforms the standard PPO algorithm on these two benchmarks across different reward models. This indicates that PF-PPO is effective in broader fields.
>
> |         | PPO with ORM | PF-PPO with ORM | PPO with Oracle | PF-PPO with Oracle | PPO with CRM | PF-PPO with CRM |
> |---------|--------------|-----------------|-----------------|--------------------|--------------|-----------------|
> | Ape210K | 84.1         | **86.2**        | 82.1            | **83.8**           | 83.9         | **84.3**        |
> | CMATH   | 92.3         | **95.1**        | 90.8            | **91.2**           | 93.1         | **94.2**        |
>
> To provide in-depth analysis on why PF-PPO works, we compare the responses generated by PF-PPO and PPO-S and provide detailed examples in Appendix C2 in the revised paper. One prominent observation we can make is that, when PF-PPO and PPO-S both generate correct codes to a given problem, the code generated by PF-PPO is shorter and more concise than that generated by PPO-S. This may benefit from the fact that PF-PPO learns from filtered samples which contain more obviously correct responses (i.e., the responses with high rewards) but less ambiguous responses (i.e., the responses with middle rewards).
>
> To provide further insights into how the obviously correct/wrong responses (with high/low rewards) and ambiguous responses (with middle rewards) look like. We provide examples of these responses to a given prompt in Appendix C1 in the revised paper. The key observation is that by filtering samples, PF-PPO can better learn to solve the problems following a standard approach. Specifically, we have the following observations: 1) The responses with low rewards typically contain obvious mistakes (such as syntax errors). 2) The responses with high rewards are typically consistent with a standard way of solution. 3) Sometimes, the responses with middle rewards try a non-standard approach to solve the problem. The reward model does not well recognize such solutions and therefore does not give a clear high reward, even if the response is correct. 4) Sometimes, the responses with middle rewards follow the standard way of solution but contain minor errors (such as wrong boundary conditions) or incorrectly mixes two correct approaches.
>
> > [Presentation] The foundation of the proposed method is the reliability measure (or accuracy) for reward model prediction. The paper propose to use the the coefficient of determination (R-squared), but the paper do not include details for computing this measure, and do not clearly state the motivation and/or why this measure is valid to measure the reliability of reward estimate. For instance, why choose this measure? why not other measures, e.g., errors of the statistical estimation? Lack of clarity hinders understanding for the audience who are not familiar with R-squared. There are many other presentation issues, which makes the paper hard to understand.
>
> According to the suggestion from the reviewer, we have updated the paper and include the details for computing the measure and the motivation. The motivation is to measure how well the rewards can be used to indicate the actual scores. Other possible metrics (such as the error of statistical estimation) are similar to R-squared. For example, the ranks of different filtering strategies we considered in our paper remain the same under the error of statistical estimation.
>
> **References**:
>
> [1] Ouyang, Long, et al. "Training language models to follow instructions with human feedback." Advances in neural information processing systems 35 (2022): 27730-27744.
>
> [2] Huang, Shengyi, et al. "The 37 implementation details of proximal policy optimization." The ICLR Blog Track 2023(2022).
>
> [3] Naik, Abhishek, et al. "Reward Centering." arXiv preprint arXiv:2405.09999 (2024).

---

> ### Comment · Reviewer_FPLk · 2024-11-26
> **Reviewer Response**
>
> Thank you for your rebuttal.
>
> My primary concern is that ***the current paper lacks depth; it is based on empirical observations but fails to uncover the underlying principles that explain these observations.*** Specifically:
>
> - **Lack of Fundamental Theories**
>
> I would like to further clarify the previous comment of Weakness 1.
>
> The authors claim that
>
> > We find that, across different sets of samples, the reward model is more reliable when it gives high or low rewards than when it gives moderate rewards.
>
> This claim would become unconvincing for the readers due to the lack of fundamental theories or rationales. Without a compelling theory to substantiate this claim, doubts remain. Indeed, without the fundamental rationales for the empirical observations, it would always remain unclear whether this property hold for certain scopes until experiments are conducted on that scope. Therefore, this absence of an underlying theoretical basis limits the contribution and significance of the empirical findings.
>
> To be more specific, the readers would be prompted to ask: what is the cause of reward models in specific domains being more reliable when they give high or low rewards? Is this property due to training methods, base models, or application domains?
>
> - **Ad-hoc Design**
>
> Due to the fact that the paper does not reveal a fundamental issue in reward model training, the proposed policy filtration strategies are designed ad-hoc in response to the observed empirical phenomena. This limits the contribution, as the readers don't fully understand what is the proper application domain for the proposed approach.
>
> **In response to the rebuttal:**
>
> I appreciate the authors conducting additional experiments on math reasoning. However,
> - I don't see results on standard english math benchmarks, such as GSM8K, MATH, College-Math, OlympiadsBench.
> - Besides, math reasoning is similar to code generation, as both are easy to achieve outcome-based verification for a golden reward. What about preference-based reward models, e.g., helpfulness, safety?
>
> If the authors could make verifiable claims about the proposed approach effective for what types of reward models (e.g., training conditions, application domains, etc.), this at least clearly delineate the scope of application for the proposed method, and I would raise the score.

---

> ### Author Response · Authors · 2024-11-26
> **Author's Response to Reviewer FPLk**
>
> Thank you for your feedback. We would like to address the following points:
>
> 1. **Why do we focus on reasoning tasks, including math tasks and code generation, instead of safety or helpfulness tasks?**
>
> The evaluation of safety or helpfulness tasks heavily relies on the assessment of GPT models, which have been demonstrated to be susceptible to various exploits as documented in several works [1, 2, 3]. As a result, we consider reasoning tasks to be more appropriate for algorithmic research in the RLHF (Reinforcement Learning with Human Feedback) area.
>
> 2. **Why do we use the preference-based reward model instead of oracle rewards?**
>
> Deepseek-Math and Qwen-Math have shown that oracle rewards are not suitable for the RLHF process, as they lead to inferior performance compared to preference-based reward models or mixed reward models that combine preference-based and oracle rewards.
>
> Due to limited time for the rebuttal period, we conducted our experiments using Chinese Benchmarks rather than English Benchmarks. We will add other benchmarks in the future revision.
>
> We hope this response clarifies our rationale for the chosen approaches and the scope of our experiments.
>
> [1] Dubois Y, Galambosi B, Liang P, et al. Length-controlled alpacaeval: A simple way to debias automatic evaluators[J]. arXiv preprint arXiv:2404.04475, 2024.
>
> [2] Chen G H, Chen S, Liu Z, et al. Humans or llms as the judge? a study on judgement biases[J]. arXiv preprint arXiv:2402.10669, 2024.
>
> [3] Zhang X, Xiong W, Chen L, et al. From lists to emojis: How format bias affects model alignment[J]. arXiv preprint arXiv:2409.11704, 2024.

---

### Official Review · Reviewer_J5Zt · 2024-11-04

**Soundness:** 2
**Presentation:** 2
**Contribution:** 1
**Rating:** 5
**Confidence:** 3

**Summary:**

The paper "Policy Filtration in RLHF to Fine-Tune LLM for Code Generation" proposes a filtering strategy for Reinforcement Learning from Human Feedback (RLHF) to improve code generation in large language models (LLMs). The authors introduce Policy Filtration for Proximal Policy Optimization (PF-PPO), a method that aims to mitigate noise from inaccurate reward models by filtering unreliable samples during training. The method is benchmarked on the HumanEval, MBPP, and a newly created LeetCode Contest dataset, demonstrating some performance gains. The paper claims that filtering improves reward model reliability, leading to better overall policy outcomes.

**Strengths:**

The paper addresses the challenge of reward model noise in RLHF, a known issue in the field. By focusing on reward reliability and noise reduction, it touches on a potentially impactful area for RL-based code generation.

**Weaknesses:**

Lack of Novelty in Approach: The proposed method, PF-PPO, builds upon existing Proximal Policy Optimization (PPO) by simply filtering samples. This approach does not introduce substantial innovation, as similar filtering and sampling techniques are well-explored in RL literature. The study could benefit from a deeper analysis of why existing filtering methods are inadequate for RLHF in code generation.

Limited Practical Impact: Although the paper demonstrates slight improvements on specific benchmarks, it does not provide a strong case for the broader utility of PF-PPO. The marginal performance gains (e.g., 0.7% to 10% across different benchmarks) may not justify the added complexity. Furthermore, focusing primarily on accuracy metrics without detailed qualitative analysis limits the assessment of how this approach improves code generation's real-world usability.

Over-reliance on Benchmark Metrics: The paper heavily emphasizes quantitative metrics without exploring the implications of these improvements on code quality or robustness. A qualitative analysis of code examples generated by PF-PPO would help clarify its value, especially given that code generation tasks require more nuanced evaluation beyond pass/fail metrics.

Omission of Real-World Considerations: The filtering strategy relies on preset thresholds and assumptions about reward distribution, which may not generalize to more diverse or complex RLHF tasks. Additionally, the computational overhead and feasibility of implementing this method in large-scale production are not discussed, raising concerns about practical applicability.

**Questions:**

Given that PF-PPO’s improvements are incremental, can the authors provide more justification for why this method is advantageous over simpler techniques like weighted sampling?

Could the authors share examples of generated code to illustrate how the proposed filtering affects code quality, readability, or maintainability beyond accuracy metrics?

How does PF-PPO’s computational complexity compare to existing methods when deployed in real-world environments, particularly for larger models and datasets?

---

> ### Author Response · Authors · 2024-11-22
> **Author's Response to Reviewer J5Zt (1/2)**
>
> We would like to thank the reviewer for valuable feedback and for the comments and questions for additional clarity. Below, we address the key comments and questions.
>
> > Lack of Novelty in Approach: The proposed method, PF-PPO, builds upon existing Proximal Policy Optimization (PPO) by simply filtering samples. This approach does not introduce substantial innovation, as similar filtering and sampling techniques are well-explored in RL literature. The study could benefit from a deeper analysis of why existing filtering methods are inadequate for RLHF in code generation.
>
> While filtering samples are studied in RL literature, previous research focuses on different purposes from ours and thus is not directly applicable to RLHF. Prioritized experience replay (PER) [1] is one of the most prominent examples of filtering sampling in RL literature, and it is used to focus on the significant samples (with large TD errors). PER does not suit the scenario with noisy feedback since the noise is easily recognized as the significance. Other follow-up techniques [2-4] filter samples in experience replay for other purposes such as data augmentation, enhancing exploration, and eliminating off-policy samples. SAUNA [5] filters out irrelevant samples to boost RL, but their algorithmic design is targeted for filtering out trajectories where the locomotion agent gets trapped. Since RLHF is a special scenario where the reward feedback contains high noise, the objective of filtering samples in our paper (to reduce noise) is novel.
>
> > Limited Practical Impact: Although the paper demonstrates slight improvements on specific benchmarks, it does not provide a strong case for the broader utility of PF-PPO.
>
> To provide more evidence on the broader utility of PF-PPO, we conducted additional experiments on math reasoning tasks and we updated the experiment results to Appendix B. Specifically, we use Qwen1.5-7B [6] as the SFT model and Ape210K [7] and CMATH [8] as the evaluation benchmarks. Other training settings (including the training datasets) are the same as [9]. We also try different reward functions (see details in our revised paper). We observe that PF-PPO consistently outperforms the standard PPO algorithm on these two benchmarks across different reward models. This indicates that PF-PPO is effective in broader fields.
>
> |         | PPO with ORM | PF-PPO with ORM | PPO with Oracle | PF-PPO with Oracle | PPO with CRM | PF-PPO with CRM |
> |---------|--------------|-----------------|-----------------|--------------------|--------------|-----------------|
> | Ape210K | 84.1         | **86.2**        | 82.1            | **83.8**           | 83.9         | **84.3**        |
> | CMATH   | 92.3         | **95.1**        | 90.8            | **91.2**           | 93.1         | **94.2**        |
>
> > The marginal performance gains (e.g., 0.7% to 10% across different benchmarks) may not justify the added complexity.
>
> We want to clarify the misunderstanding from the reviewer that our algorithm is associated with “added complexity”. Actually, PF-PPO improves the performance with less computational cost (see the analysis in Table 4) and the performance comparison is made with the same amount of response generation. (PF-PPO actually masks ambiguous samples.) Despite decreased computation, PF-PPO achieves positive improvement across various benchmarks, which well demonstrates the effectiveness of PF-PPO.

---

> ### Author Response · Authors · 2024-11-22
> **Author's Response to Reviewer J5Zt (2/2)**
>
> > Furthermore, focusing primarily on accuracy metrics without detailed qualitative analysis limits the assessment of how this approach improves code generation's real-world usability.
> > Over-reliance on Benchmark Metrics: The paper heavily emphasizes quantitative metrics without exploring the implications of these improvements on code quality or robustness. A qualitative analysis of code examples generated by PF-PPO would help clarify its value, especially given that code generation tasks require more nuanced evaluation beyond pass/fail metrics.
>
> To assess PF-PPO from a qualitative perspective, we compare the responses generated by PF-PPO and PPO-S and provide detailed examples in Appendix C2 in the revised paper. One prominent observation we can make is that, when PF-PPO and PPO-S both generate correct codes to a given problem, the code generated by PF-PPO is shorter and more concise than that generated by PPO-S. This may benefit from the fact that PF-PPO learns from filtered samples which contain more obviously correct responses (i.e., the responses with high rewards) but less ambiguous responses (i.e., the responses with middle rewards).
> To provide further insights into how the obviously correct/wrong responses (with high/low rewards) and ambiguous responses (with middle rewards) look like. We provide examples of these responses to a given prompt in Appendix C1 in the revised paper. The key observation is that by filtering samples, PF-PPO can better learn to solve the problems following a standard approach. Specifically, we have the following observations: 1) The responses with low rewards typically contain obvious mistakes (such as syntax errors). 2) The responses with high rewards are typically consistent with a standard way of solution. 3) Sometimes, the responses with middle rewards try a non-standard approach to solve the problem. The reward model does not well recognize such a solution and therefore does not give a high reward, even if the response is correct. 4) Sometimes, the responses with middle rewards follow the standard way of solution but contain minor errors (such as wrong boundary conditions).
>
> > Omission of Real-World Considerations: The filtering strategy relies on preset thresholds and assumptions about reward distribution, which may not generalize to more diverse or complex RLHF tasks. Additionally, the computational overhead and feasibility of implementing this method in large-scale production are not discussed, raising concerns about practical applicability.
>
> Although a specific filtering strategy relies on the thresholds and the assumption about reward distribution, the proposed procedure (using $R^2$) to select from candidate filtering strategies aims at making our approach more generalizable to other RLHF tasks. Actually, following this procedure, PF-PPO can be adopted successfully to math reasoning tasks (see Appendix B in the revised paper), providing a promising signal on the generalizability of PF-PPO to other RLHF tasks.
> We need to clarify that PF-PPO improves the performance with less computational cost (see the analysis in Table 4) and the performance comparison is made with the same amount of response generation. (PF-PPO actually masks ambiguous samples.) The only computational overhead may come from the procedure to select the filtering strategies. This procedure only involves collecting samples from the SFT policy, querying their actual scores, and performing simple statistics for different strategies on this set of samples. However, this overhead is constant and does not scale with RLHF training.
>
> **References**:
>
> [1] Horgan, Dan, et al. "Distributed prioritized experience replay." arXiv preprint arXiv:1803.00933 (2018).
>
> [2] Novati, Guido, and Petros Koumoutsakos. "Remember and forget for experience replay." International Conference on Machine Learning. PMLR, 2019.
>
> [3] Andrychowicz, Marcin, et al. "Hindsight experience replay." Advances in neural information processing systems 30 (2017).
>
> [4] Liu, Hao, et al. "Competitive experience replay." arXiv preprint arXiv:1902.00528 (2019).
>
> [5] Flet-Berliac, Yannis, and Philippe Preux. "Only relevant information matters: Filtering out noisy samples to boost rl." arXiv preprint arXiv:1904.04025 (2019).
>
> [6] Qwen Team. “Introducing Qwen1.5”, 2024
>
> [7] Zhao, Wei, et al. "Ape210k: A large-scale and template-rich dataset of math word problems." arXiv preprint arXiv:2009.11506 (2020)
>
> [8] Wei, Tianwen, et al. "Cmath: Can your language model pass chinese elementary school math test?." arXiv preprint arXiv:2306.16636 (2023).
>
> [9] Zhou J, Jiang C, Shen W, et al. Leveraging Web-Crawled Data for High-Quality Fine-Tuning[C]//Findings of the Association for Computational Linguistics: EMNLP 2024. 2024: 11297-11312.

---

### Official Review · Reviewer_n6hr · 2024-11-04

**Soundness:** 3
**Presentation:** 1
**Contribution:** 3
**Rating:** 6
**Confidence:** 2

**Summary:**

The paper introduces a new way of filtering model generated samples during the PPO training stage that the authors call Policy Filtration PPO (PF-PPO). The method changes the sampling distribution of the target LLM by post-hoc selecting the generated samples using the reward model, via a heuristic that filters for generated samples that are likely to have reward values that correlate highly (measured by R^2) with ground truth accuracy. In particular, the authors found that selecting the generated samples with high/low valued rewards improve the final target LLM’s code generation performance. The authors show that PF-PPO is able to achieve SOTA result on code generation in 7B class of open source LLM models.

**Strengths:**

1. While data selection during reward modeling has previously been discussed, the paper addresses a novel issue of post-hoc data selection during PPO training after RM has been trained. The authors found simple yet effective heuristic of choosing samples that are unlikely to have incorrect reward signal.
2. The empirical results shown in the paper, given the simplicity of the data selection method, is impressive. I could see this kind of method be widely adopted in RL.

**Weaknesses:**

1. The paper broadly suffers from writing clarity, and has the tendency to overcomplicate concepts that appear to have simpler alternative explanations.  For example, the formulation of policy filtration could be drastically simplified. If I understand the method correctly, the algorithm is in essence sampling model generated responses by some heuristics which maximizes the R^2 between reward value and ground truth accuracy. The discussion around parameterized sampling, straight through estimator, seem redundant and was confusing. It is not clear how the “actual score” for code generation task was obtained and the appendix section seems straight from the template.
2. Some key discussion seem to be missing that limit the general applicability of the paper. The paper discusses application in code generation domain, where there exist oracle reward modeling (via code execution). As such, using R^2 to find the samples with the least RM noise seem redundant (why not just use the oracle RM)? Where this question seems much more pertinent is in tasks that don't have objective ground truths? What if the "actual score" is noisy?

**Questions:**

I strongly recommend the reviewer to significantly reduce the unnecessary discussions and elaborations in section 3/4 and focus their attention on
1. analysis of the impact of the algorithm. In particular, insights into what data are selected and discarded as a result of the proposed algorithm, and their consequent impacts on the RLHF procedure, will be highly useful for RLHF practitioners.
2. generalization of the proposed algorithm to other problem domains (e.g. other tasks with objective ground truth such as mathematical reasoning, and tasks where such objective ground truth do not exist)

---

> ### Author Response · Authors · 2024-11-22
> **Author's Response to Reviewer n6hr (1/2)**
>
> We would like to thank the reviewer for valuable feedback and for the comments and questions for additional clarity. Below, we address the key comments and questions.
>
> > The paper broadly suffers from writing clarity, and has the tendency to overcomplicate concepts that appear to have simpler alternative explanations. For example, the formulation of policy filtration could be drastically simplified. If I understand the method correctly, the algorithm is in essence sampling model generated responses by some heuristics which maximizes the R^2 between reward value and ground truth accuracy. The discussion around parameterized sampling, straight through estimator, seem redundant and was confusing.
>
> We thank the reviewer for the valuable feedback and we have revised the paper to simplify the description of our algorithm according to the suggestion for the reviewer (see the “Training objective” part in Section 4). Specifically, we removed the description related to straight-through estimator (STE) which was used to justify the algorithmic design but was somewhat redundant and confusing. We kept the description of parameterized sampling (involving, e.g., weight vectors) since it is necessary to formally describe the process and introduce different variants of PF-PPO (BoN, BR, BW) more clearly.
>
> > It is not clear how the “actual score” for code generation task was obtained and the appendix section seems straight from the template.
>
> The actual score is obtained by feeding the generated code to a code evaluator. The code evaluator runs the code on several test cases and checks whether all the answers are correct. We have revised the appendix section.
>
> > Some key discussion seem to be missing that limit the general applicability of the paper. The paper discusses application in code generation domain, where there exist oracle reward modeling (via code execution). As such, using R^2 to find the samples with the least RM noise seem redundant (why not just use the oracle RM)? Where this question seems much more pertinent is in tasks that don't have objective ground truths? What if the "actual score" is noisy?
>
> Directly using the oracle reward model (i.e., the actual score) for training is not a good choice for code generation since the oracle reward is sparse. Therefore, our approach is not redundant. To illustrate this point, we conducted an experiment to run PPO-S(tandard) using the oracle reward. We show its performance on the LeetCode benchmark in the table below.
>
> |          | PPO-S + Oracle | PPO-S | PF-PPO (BR) |   |
> |----------|----------------|-------|-------------|---|
> | LeetCode | 23.8           | 25.2  | 33.0        |   |
>
> We observe that PPO-S trained with the oracle reward model is not as good as PPO-S and PF-PPO (BR) trained using a learned reward model. Further, we observe that, by using oracle reward, the policy learns well on simple problems but  struggles on hard problems. This is because the reward model can provide a gradient by comparing two incorrect responses with different qualities. Besides, querying from an oracle is more time-consuming than querying the reward model. The similar conclusion on using the oracle reward model can also be drawn from the experimental results provided in the answer to the last question from the reviewer (see below).
>
> Further explanation: Using the reward model faces a tradeoff between its generalization/reward shaping ability and the noise, and our method serves as a way to reduce the noise induced by the reward model while keeping its generalization.
>
> For tasks that don’t have ground truths, our method still applies. In general, we do have at least a noisy way to compare between different responses or different policies. In this case, we can still score a set of responses generated by the SFT policy and try different filtration strategies to see which strategy leads to the best alignment ($R^2$) between the reward and the actual score. When the actual score is noisy, the overall alignment may decrease, but it should not change the rank of different strategies so we can still be able to select a good strategy.

---

> ### Author Response · Authors · 2024-11-22
> **Author's Response to Reviewer n6hr (2/2)**
>
> > analysis of the impact of the algorithm. In particular, insights into what data are selected and discarded as a result of the proposed algorithm, and their consequent impacts on the RLHF procedure, will be highly useful for RLHF practitioners.
>
> To analyze the impact of PF-PPO, we compare the responses generated by PF-PPO and PPO-S and provide detailed examples in Appendix C2 in the revised paper. One prominent observation we can make is that, when PF-PPO and PPO-S both generate correct codes to a given problem, the code generated by PF-PPO is shorter and more concise than that generated by PPO-S. This may benefit from the fact that PF-PPO learns from filtered samples which contain more obviously correct responses (i.e., the responses with high rewards) but less ambiguous responses (i.e., the responses with middle rewards).
>
> To provide insights into how the obviously correct/wrong responses (with high/low rewards) and ambiguous responses (with middle rewards) look like. We provide examples of these responses to a given prompt in Appendix C1 in the revised paper. The key observation is that by filtering samples, PF-PPO can better learn to solve the problems following a standard approach. Specifically, we have the following observations: 1) The responses with low rewards typically contain obvious mistakes (such as syntax errors). 2) The responses with high rewards are typically consistent with a standard way of solution. 3) Sometimes, the responses with middle rewards try a non-standard approach to solve the problem. The reward model does not well recognize such a solution and therefore does not give a high reward, even if the response is correct. 4) Sometimes, the responses with middle rewards follow the standard way of solution but contain minor errors (such as wrong boundary conditions).
>
> > generalization of the proposed algorithm to other problem domains (e.g. other tasks with objective ground truth such as mathematical reasoning, and tasks where such objective ground truth do not exist)
>
> We thank the reviewer for the valuable suggestion. To see whether PF-PPO can generalize to other domains, we applied PF-PPO to solve math problems. Specifically, we use Qwen1.5-7B [1] as the SFT model and Ape210K [2] and CMATH [3] as the evaluation benchmarks. Other training settings (including the training datasets) are the same as [4].
> We use three types of reward models: the original reward model (ORM) that is trained on preference datasets using a Bradley–Terry model, an oracle model (Oracle) that extracts the final answer from the response and compares it with the ground truth, and a combined reward model (CRM) that integrates the above two models (cf. Qwen-Math [5] and Deepseek-Math [6]). Furthermore, we compare our methods to the standard PPO (PPO-S) using these reward models.
>
> |         | PPO with ORM | PF-PPO with ORM | PPO with Oracle | PF-PPO with Oracle | PPO with CRM | PF-PPO with CRM |
> |---------|--------------|-----------------|-----------------|--------------------|--------------|-----------------|
> | Ape210K | 84.1         | **86.2**        | 82.1            | **83.8**           | 83.9         | **84.3**        |
> | CMATH   | 92.3         | **95.1**        | 90.8            | **91.2**           | 93.1         | **94.2**        |
>
> We observe that PF-PPO consistently outperforms the PPO algorithm on these two benchmarks across different reward models. This indicates that PF-PPO is effective in broader fields.
> In addition, the experiment results indicate that even if we can have access to the ground truth, using the oracle as the reward function does not perform as well as using a reward model (either the original reward model or the combined model).
>
> **References**:
>
> [1] Qwen Team. “Introducing Qwen1.5”, 2024
>
> [2] Zhao, Wei, et al. "Ape210k: A large-scale and template-rich dataset of math word problems." arXiv preprint arXiv:2009.11506 (2020)
>
> [3] Wei, Tianwen, et al. "Cmath: Can your language model pass chinese elementary school math test?." arXiv preprint arXiv:2306.16636 (2023).
>
> [4] Zhou J, Jiang C, Shen W, et al. Leveraging Web-Crawled Data for High-Quality Fine-Tuning[C]//Findings of the Association for Computational Linguistics: EMNLP 2024. 2024: 11297-11312.
>
> [5] Yang A, Zhang B, Hui B, et al. Qwen2. 5-math technical report: Toward mathematical expert model via self-improvement[J]. arXiv preprint arXiv:2409.12122, 2024.
>
> [6] Shao Z, Wang P, Zhu Q, et al. Deepseekmath: Pushing the limits of mathematical reasoning in open language models[J]. arXiv preprint arXiv:2402.03300, 2024.

---

> > ### Comment · Reviewer_n6hr · 2024-11-24
> > **Thank you**
> >
> > In this revision, the authors significantly revised their manuscript and
> > 1. rewrote the methods section to improve clarity
> > 2. added analysis of data selection by PF-PPO
> > 3. added additional results for math reasoning.
> >
> > The authors have addressed all of my concerns, hence raising my previous score from 5 to 6.

---

### Author Response · Authors · 2024-11-22

We would like to thank all the reviewers for the valuable feedback. According to reviewers’ feedback, we have revised and updated our paper. The revised parts are highlighted in blue. Specifically, we made the following major updates:
- (Appendix A) Our algorithm is motivated by the observation that the reward model may be unreliable when it gives middle rewards. To verify the universality of this property, we analyzed the reward model on other two benchmarks.
- (Appendix B) We adopted our algorithm PF-PPO to the math reasoning tasks and observed that PF-PPO improves the performance on different benchmarks using different reward functions.
- (Appendix C) We provided qualitative results on 1) how responses with high/middle/low rewards look like and why responses with middle rewards are unreliable; and 2) the qualitative difference between the code generated by our algorithm PF-PPO and standard PPO.

We sincerely ask the reviewers to re-evaluate our paper based on our rebuttal and the revised version of the paper.

---

### Meta-Review · Area_Chair_qMCt · 2024-12-20

**Metareview:**

This paper proposes to filter low-quality reward samples used to train intermediate reward models to improve performance in RLHF. While the paper appears to be tackling an interesting and important problem, the reviewers had concerns about the ad-hoc nature of the proposed approach as well as the limited scope of the experiments. Significant revision is needed to improve the paper before acceptance might be considered.

**Additional Comments On Reviewer Discussion:**

The authors made some progress in addressing the concerns of the reviewers. However, fundamental concerns about the ad-hoc nature of the proposed solution remain, as well as concerns about the scope of the experiments. The authors are encouraged to pursue either more principled methodologies or expand the scope of their experiments to improve confidence in the ad-hoc technique they proposed.

---

### Decision · Program_Chairs · 2025-01-22

Reject